# Bioactive Compounds and Evaluation of Antioxidant, Cytotoxic and Cytoprotective Effects of Murici Pulp Extracts (*Byrsonima crassifolia*) Obtained by Supercritical Extraction in HepG2 Cells Treated with H_2_O_2_

**DOI:** 10.3390/foods10040737

**Published:** 2021-03-30

**Authors:** Flávia Cristina Seabra Pires, Joicy Corrêa de Oliveira, Eduardo Gama Ortiz Menezes, Ana Paula de Souza e Silva, Maria Caroline Rodrigues Ferreira, Leticia Maria Martins Siqueira, Andryo Orfi Almada-Vilhena, Julio Cesar Pieczarka, Cleusa Yoshiko Nagamachi, Raul Nunes de Carvalho Junior

**Affiliations:** 1LABEX (Extraction Laboratory), LABTECS (Supercritical Technology Laboratory), PPGCTA (PostGraduate Program in Food Science and Technology), ITEC (Institute of Technology), UFPA (Federal University of Pará), Augusto Corrêa Street S/N, Guamá, Belém, PA 66075-900, Brazil; flaviapiress@gmail.com (F.C.S.P.); anapaula-eng@hotmail.com (A.P.d.S.eS.); carolinerof@gmail.com (M.C.R.F.); 2LABEX (Extraction Laboratory), FEA (College of Food Engineering), ITEC (Institute of Technology), UFPA (Federal University of Pará), Augusto Corrêa Street S/N, Guamá, Belém, PA 66075-900, Brazil; joicyo90@gmail.com; 3LABEX (Extraction Laboratory), PRODERNA (Postgraduate Program in Natural Resources Engineering in the Amazon), ITEC (Institute of Technology), UFPA (Federal University of Pará), Augusto Corrêa Street S/N, Guamá, Belém, PA 66075-900, Brazil; ortizegom@hotmail.com (E.G.O.M.); leticiammsiqueira@outlook.pt (L.M.M.S.); 4CEABIO (Center for Advanced Studies of the Biodiversity and Cell Culture Laboratory), PCT-Guamá (Guamá Science and Technology Park), UFPA (Federal University of Pará), Augusto Corrêa Street S/N, Guamá, Belém, PA 66075-900, Brazil; andryoorfi@hotmail.com (A.O.A.-V.); julio@ufpa.br (J.C.P.); cleusa@ufpa.br (C.Y.N.); 5LABEX (Extraction Laboratory), LABTECS (Supercritical Technology Laboratory), FEA (College of Food Engineering), ITEC (Institute of Technology), UFPA (Federal University of Pará), Augusto Corrêa Street S/N, Guamá, Belém, PA 66075-900, Brazil

**Keywords:** supercritical CO_2_, supercritical CO_2_+ethanol, global yield, lutein, phenolic compounds, flavonoids, fatty acids, triglycerides, ORAC, DPPH

## Abstract

The use of clean technologies in the development of bioactive plant extracts has been encouraged, but it is necessary to verify the cytotoxicity and cytoprotection for food and pharmaceutical applications. Therefore, the objective of this work was to obtain the experimental data of the supercritical sequential extraction of murici pulp, to determine the main bioactive compounds obtained and to evaluate the possible cytotoxicity and cytoprotection of the extracts in models of HepG2 cells treated with H_2_O_2_. The murici pulp was subjected to sequential extraction with supercritical CO_2_ and CO_2_+ethanol, at 343.15 K, and 22, 32, and 49 MPa. Higher extraction yields were obtained at 49 MPa. The oil presented lutein (224.77 µg/g), oleic, palmitic, and linoleic, as the main fatty acids, and POLi (17.63%), POO (15.84%), PPO (13.63%), and LiOO (10.26%), as the main triglycerides. The ethanolic extract presented lutein (242.16 µg/g), phenolic compounds (20.63 mg GAE/g), and flavonoids (0.65 mg QE/g). The ethanolic extract showed greater antioxidant activity (122.61 and 17.14 µmol TE/g) than oil (43.48 and 6.04 µmol TE/g). Both extracts did not show cytotoxicity and only murici oil showed a cytoprotective effect. Despite this, the results qualify both extracts for food/pharmaceutical applications.

## 1. Introduction

Murici (*Byrsonima crassifolia*) originates from Central and South Americas, where a tropical and temperate climate prevails. The fruit has been used for production of pulps, ice creams, etc. [1]. It has also been used in folk medicine, since the pre-Hispanic era, due to its therapeutic effects attributed to its bioactive compounds, such as antihyperglycemic, antihyperlipidemic, and antioxidant activities [2,3].

About these compounds contained in murici pulp, some studies have evaluated their phytochemical profile. In the carotenoid class, Rodrigues et al. [4] identified as major carotenoids (all-E)-lutein, (all-E)-zeaxanthin, and (all-E)-lutein-3-O-myristate coeluted with (all-E)-β-carotene, where the free and esterified lutein corresponded to 48% total carotenoid content. In the study by Irías-Mata et al. [5], the main carotenoids were (all-E)-lutein, (all-E)-zeaxanthin, and (all-E)-β-carotene, where (all-E)-lutein contributed 80–89% of total carotenoids. In the study carried out by Mariutti et al. [6], the majorities were (all-E)-lutein and (all-E)-zeaxanthin, where lutein represented about 67.39% of total carotenoids.

Another class that stands out is the phenolic compounds, where in the study developed by Gordon et al. [7] quercetin hexoside, quercetin and tetragalloylquinic acid were identified as the main carotenoids in murici pulp. In the work of Mariutti et al. [3], the major phenolic compounds were quercetin, gallic acid, quercetin hexoside, and quercetin pentoside, where quercetin represented about 66% of total phenolic compounds. Among the unsaturated fatty acids present in murici oil, the main compounds identified were oleic, linoleic, and linolenic acids, which make up about 64% of the lipid profile [8].

Due to the presence of several bioactive compounds in plants, its use has been increasingly explored due to the numerous benefits of its use on human health. One of the ways to verticalize the consumption of plant products is extract manufacturing. Some studies have addressed the obtaining of murici pulp extracts obtained by different extraction techniques and solvents such as exhaustive maceration with acetone [6], agitation and sonication on ultrasound with a methanol/ethyl acetate/petroleum ether extraction solution (1:1:1, *v*/*v*/*v*) [5], consecutive centrifugation with methanol extraction solution at 50% (*v*/*v*), acetone extraction solution at 70% (*v*/*v*), and water [9], agitation, and centrifugation with methanol/water extracting solution (8:2, *v*/*v*) [3], among others.

It is important to note that in recent years, there has been a growth in global interest in the use of clean technologies for the development of food and pharmaceutical products. Therefore, the use of supercritical technology appears as a potential alternative to obtain 100% pure plant extracts, free of solvents, and environmentally “green” [10,11,12], where in the scientific literature can be found two works that addressed the extraction of murici pulp by extraction The work developed by Santos et al. [8] addressed the quality parameters and thermogravimetric and oxidative profile of murici oil (*Byrsonima crassifolia* L.) obtained by supercritical CO_2_, and the work developed by Pires et al. [13] addresses the determination of process parameters and bioactive properties of the murici pulp (*Byrsonima crassifolia*) extracts obtained by supercritical extraction, the latter being the scientific basis of the present study.

Many studies have approached the use of supercritical CO_2_ to obtain nonpolar plant extracts. Among the main compounds present in such extracts, unsaturated fatty acids and carotenoids stand out due to their beneficial health effects [14,15,16,17]. In addition, other studies have addressed their applications due to the characteristics of their medium and long-chain triglycerides [18,19,20,21]. Thus, it is important to know the type of triglyceride present in the extracts, since the prediction through the composition of fatty acids have showed a triglyceride profile quite similar to that obtained experimentally [22].

Although extraction with supercritical CO_2_ is used on large scale, some studies have found that the residual extraction bed concentrates polar bioactive compounds, which allows defatted pulps to be better utilized in the industry, and minimizes the waste of high-value industrial residues in the environment [23,24,25]. For polarity modification, it is necessary to use a co-solvent, such as ethanol, because of its GRAS status, generally recognized as safe for use in food and pharmaceutical products, in small quantities [26].

Due to the potential application of plant extracts in the food and pharmaceutical area, several studies have been developed with the objective of verifying the possible cytotoxicity and cytoprotection of these extracts in cell models, where the human hepatoma cell line (HepG2) has been widely applied [27,28,29,30]. This strain has been used for demonstrating to be more sensitive to cytotoxic compounds compared to other cell lines (HeLa, ECC-1 and CHO-K1). In addition, HepG2 cells exhibit genotypic and phenotypic characteristics of normal liver cells, allowing for a wide variety of liver-specific metabolic responses, thus increasing the likelihood of predicting human susceptibility to the biological effects of plant extracts [30,31,32,33].

Therefore, the objective of this work was to obtain the experimental data of the supercritical sequential extraction of murici pulp, to determine the main bioactive compounds obtained and to evaluate the possible cytotoxicity and cytoprotection of the extracts in models of HepG2 cells treated with H_2_O_2_.

## 2. Materials and Methods

### 2.1. Raw Materials and Sample Preparation

Murici pulp was obtained in Terra Alta town (Pará, Brazil) (01°02′25.9″ S and 47°54′12.3″ W) (Accession Number: 199530/ Barcode: IAN199530/ Sisgen: A3DB8EB). Approximately 30 kg of pulp were lyophilized in a semi-industrial lyophilizer (model LJI 015, JJ Científica, São Carlos, Brazil), at 233.15 K, for two days. The freeze-dried pulp obtained presented 5.46 ± 0.30 g/100 g of moisture, 2.02 ± 0.02 g/100 g of ashes, 0.55 ± 0.02 g/100 g of proteins, 10.02 ± 0.15 g/100 g of lipids, 21.60 ± 0.59 g/100 g of sugars, 56.42 ± 0.21 μg/g (d.b.) of lutein, 15.43 ± 0.02 mg GAE/g (d.b.) of phenolic compounds, and 6.62 ± 0.36 μmol TE/g (d.b.) of antioxidant capacity (ORAC) [13]. The freeze-dried pulp was vacuum-packed, protected from light, and stored under freezing until extraction (253.15 K ± 1 K).

### 2.2. Sequential Supercritical Extractions: CO_2_–SFE (Extraction with Supercritical CO_2_) and CO_2_+EtOH–SFE (Extraction with Supercritical CO_2_ and Ethanol)

The sequential supercritical extractions were performed in the Extraction Laboratory (LABEX/UFPA/Brazil) in a SPE-ED SFE unit (model 7071, Applied Separations, Allentown, PA, USA), coupled to a CO_2_ cylinder, a compressor (model CSA 78, Schulz S/A, Joinville, Brazil), a recirculator (model F08400796, Polyscience, Niles, IL, USA), and a CO_2_ flowmeter (model M 5SLPM, Alicat Scientific, New York, USA). Based on the results of the evaluation of the solvent flow rates over the dynamic period of CO_2_-SFE in the supercritical extraction of murici pulp, published in an article by our research group [13], the global yield isotherms of the present work were obtained in 343.15 K, under CO_2_ densities of approximately 700, 800, and 900 kg/m^3^, and CO_2_+Ethanol densities of approximately 775, 858, and 944 kg/m^3^. For the calculation of these densities at this temperature, pressures of 22, 32, and 49 MPa were considered, using the Aspen Hysys software (Aspen One 8.6), which applies the cubic state equation of Peng-Robinson [34] with binary interaction parameters zero.

The beds were formed by 0.02 kg of sample, corresponding to a bed height of 0.078 m, with porosity of 0.7. The supercritical extraction process was performed in a static period (closed system) of 1800 s, and a dynamic period (open system) of 3600 s, at a flow rate of 1.33 × 10^−4^ kg/s. In the first stage, the extractions of the freeze-dried pulp with supercritical CO_2_ (CO_2_-SFE) were carried out in order to obtain the most apolar extracts (Oils), and the defatted freeze dried murici pulp. In the second step, the defatted pulps were extracted with supercritical CO_2_ and ethanol (CO_2_+EtOH-SFE) (90:10, *v*/*v*) to obtain the most polar extracts (Ethanolic extracts). The co-solvent feeding was carried out only in the static period. The operating conditions of CO_2_-SFE and CO_2_+EtOH-SFE are shown in Table 1. After these procedures, the global yields were calculated on a dry basis (d.b.), from the mathematical ratio between the extract mass and the dry sample mass (freeze dried pulp or defatted pulps). For ethanolic extracts, the residual solvents were evaporated in a CentriVap centrifuge (model 78100, Labconco, Kansas, EUA), under vacuum, at 313.15 K. The determinations were performed in triplicate and the results were expressed in in percentage on dry basis (% d.b.).

### 2.3. Characterization of the Defatted Pulps and Extracts

#### 2.3.1. Lutein Content

The lutein levels of the defatted pulps, oils and ethanolic extracts were determined according to Rodriguez-Amaya and Kimura method [35]. Acetone P.A. was used in the extraction, while a mixture of 50% ethyl ether and 50% petroleum ether (*v*/*v*) was used in the partition. The readings were made in a spectrophotometer (model IL-592, Kasuaki, Araucária, Brazil), at 445 nm. Further, the lutein absorption coefficient in ethanol (2550) was used. The analyses were performed in triplicate and the results were expressed in microgram of lutein per gram of sample on a dry basis (µg/g d.b.).

#### 2.3.2. Phenolic Compounds

The phenolic compounds content of the defatted pulps and ethanolic extracts was determined by Folin–Ciocalteu method [36,37]. About 1.00 × 10^−4^ kg of sample was subjected to extraction (1:19) with 1.90 × 10^−6^ m^3^ of an acidified ethanolic solution (70% ethanol, 29.5% distilled water and 0.5% acetic acid) (*m*/*v*). For the reactions, 1.00 × 10^−8^ m^3^ of the extracted samples were diluted 1.59 × 10^−6^ m^3^ of distilled water (1:159) (*v*/*v*). The reactions were performed using 5.00 × 10^−7^ m^3^ of diluted sample, 1.25 × 10^−5^ m^3^ of sodium carbonate solution at 7.5% and 2.50 × 10^−7^ m^3^ of Folin at 1 N (*v*/*v*/*v*). The readings were performed in a spectrophotometer (model IL−592, Kasuaki, Araucária, Brazil), at 750 nm. The standard curve of gallic acid was made starting from a 0.50 kg/m^3^ (*m*/*v*) stock solution, where five concentration points were used (0.9 to 8.3 mg/L), according to the equation of a straight line y = 0.0994x + 0.0108, where y is the absorbance and x is the concentration (See Appendix A). The determinations were performed in triplicate and the results were expressed in milligram of gallic acid equivalent per gram of sample on a dry basis (mg GAE/g d.b.).

#### 2.3.3. Flavonoid Content

The determination of total flavonoid content of the defatted pulps and ethanolic extracts was performed by the spectrophotometric method [38,39]. About 1.00 × 10^−4^ kg of sample was subjected to extraction (1:19) with 1.90 × 10^−6^ m^3^ of an acidified ethanolic solution (70% ethanol, 29.5% distilled water and 0.5% acetic acid) (*m*/*v*). For the reactions, 3.00 × 10^−7^ m^3^ of the extracted samples were diluted 2.70 × 10^−6^ m^3^ of distilled water (1:9) (*v*/*v*). The reactions were performed using 1.00 × 10^−6^ m^3^ of diluted sample and 1.00 × 10^−6^ m^3^ of aluminum chloride ethanolic solution at 2% (*v*/*v*). The readings were made in a spectrophotometer (model IL-592, Kasuaki, Araucária, Brazil), at 430 nm. The standard quercetin curve was made from a 0.10 kg/m^3^ (*m*/*v*) ethanolic stock solution, where nine concentration points were used (0.2 to 13 mg/L), according to the equation of a straight line y = 0.0715x + 0.0689, where y is the absorbance and x is the concentration (See Appendix B). The determinations were performed in triplicate and the results were expressed in milligram of quercetin equivalent per gram of sample on a dry basis (mg QE/g d.b.).

#### 2.3.4. Fatty Acids and Functional Potential

The fatty acid profile of the oils was determined by Gas Chromatography (model GC-2010, Shimadzu, Tokyo, Japan). The methylation was carried out according to the method Ce-2-66 of AOCS [40]. About 2.00 × 10^−5^ kg of murici oil was saponified with 4.00 × 10^−7^ m^3^ of methanolic sodium hydroxide solution at 0.5 N, which was subjected to a heating block at 373.15 K for 300 s. The mixture was cooled and esterified with addition of 4.00 × 10^−7^ m^3^ of boron trifluoride methanolic solution at 14%, being submitted again to heating at 373.15 K for 300 s. Then, the mixture was cooled and 8.50 × 10^−6^ m^3^ of saturated sodium chloride solution and 1.00 × 10^−6^ m^3^ of n-heptane UV-HPLC 99% were added. The mixture was stirred and left to rest until complete phase separation, where the supernatant was analyzed. The operating conditions were: helium as carrier gas, with 8.33 × 10^−7^ m^3^/s flow rate, FID detector at 523.15 K, injector with 1:100 split ratio, and injection volume of 1.00 × 10^−9^ m^3^. The column programmed temperature (TG-WAX MS A/30 m × 3.20 × 10^−4^ m × 2.50 × 10^−7^ m) was 323.15 K during 300 s, with a subsequent increase to 523.15 K. Fatty acids individual peaks were identified by comparing their retention times with known fatty acid patterns (Nu-Check-Prep, Inc., Elysian, MN, USA), under the same operating conditions. The retention times and each peak area were calculated using GC Software Solution. The determinations were done in triplicate and the results were expressed in percentage on dry basis (% d.b.). Functional potential was evaluated by atherogenicity (AI), thrombogenicity (TI), and hypocholesterolemic (HI) indexes, calculated from the polyunsaturated/saturated ratio of fatty acids [41,42].

#### 2.3.5. Probable Triglyceride Compositions

By definition, triglycerides are made up of one molecule of glycerol and three molecules of fatty acids. In the process of determining the composition of fatty acids, the oil is transesterified and the triglyceride molecules are broken down allowing the release of fatty acids [43]. Thus, the possible composition of triglycerides can be predicted from the experimental composition of fatty acids using knowledge about combinatorial analysis. The triglyceride composition of murici oil was predicted based on the statistical technique of distribution of random variables (random theory 1,2,3) proposed by Norris and Mattil [44], which applies the knowledge of combinatorial analysis associated with an algorithm, which uses the concept of simple binomial tree. In other words, theory 1, 2, and 3 assumes that each fatty acid can be randomly distributed in any of the three positions of the glycerol molecule, without distinguishing the isomers and without mischaracterizing the concept of the triglyceride molecule. According to this theory, it is considered that from an amount of X of fatty acids, it is possible to obtain an amount of X^3^ of triglycerides that will be formed. After obtaining the result of the possible combinations, each group was classified according to the theory of casual distribution through the carbon equivalent number (EC), calculated by the relationship between the carbon number (N°c) and the number of double bonds (N°db), disregarding the groups that presented concentration below 1% (EC = N°c − 2.N°db). The Visual Basic for Applications in Excel (VBA) spreadsheet used in the calculations was developed by the Laboratory of Separation Processes and Applied Thermodynamics (TERM@/UFPA).

#### 2.3.6. Oxygen Radical Absorbance Capacity (ORAC)

The antioxidant capacities of the defatted pulps, oils, and ethanolic extracts were determined by ORAC method, according to Lai et al. [45]. For the extraction, about 2.00 × 10^−4^ kg of sample was homogenized in 1.80 × 10^−6^ m^3^ of acetone P.A. (1:9) (*m*/*v*). For each sample, three dilutions were made in phosphate buffer solution pH 7.4, 800, 1200, and 1600× for oils, 60, 120, and 240× for defatted pulps and 1000, 2000, and 3000× for ethanolic extracts. About 2.50 × 10^−8^ m^3^ of each dilution was added in 96-well microplates. The standard Trolox curve was made from a 0.10 mol/m^3^ (mol/v) stock solution, where five concentrations points were used (0 to 8 µmol TE/L), according to the equation of a straight line y = 2.5162x + 1.0071, where y is the absorbance and x is the concentration (see Appendix C). A fluorescence solution in phosphate buffer solution pH 7.4 at 1.04 × 10^−8^ kg/m^3^ (*m*/*v*) was used. The readings were performed on a Fluorescence Microplate Reader (model FLx800, BioTek, Winooski, VT, USA), monitoring the effect of the samples on the fluorescence decomposition, resulting from the oxidations induced by the peroxyl radical (ROO*), produced through the thermal decomposition (310.15 K) of the AAPH solution (2.2′-Azobis (2-amidinopropane) dihydrochloride) in phosphate buffer solution pH 7.4 at 41.40 kg/m^3^ (*m*/*v*), in the presence of oxygen. The analysis was done in triplicate. The results were expressed in micromol of Trolox equivalent per gram of sample on a dry basis (µmol TE/g d.b.).

#### 2.3.7. Antioxidant Activity (DPPH)

The defatted pulps, oils, and ethanolic extracts were subjected to the determination of antioxidant activity by DPPH (2,2-diphenyl-1-picryl-hydrazyl-hidrate) spectrophotometric method, according to Brand-Williams et al. [46], with modifications. Acetone P.A. was used as extractive solvent, where about 2.00 × 10^−4^ kg of sample was homogenized in 1.80 × 10^−6^ m^3^ of acetone P.A. (1:9) (*m*/*v*). The procedure was carried out on a spectrophotometer (model IL-592, Kasuaki, Araucária, Brazil), at 517 nm, with readings monitored every 300 s until the reaction reached a plateau in 1800 s. The standard Trolox curve was performed starting from a 5 mol/m^3^ (mol/v) stock solution, where six concentrations were used (0 to 800 µM), according to the equation of a straight line y = 0.0006x + 0.0033, where y is the absorbance and x is the concentration (See Appendix D). The reactions were performed using 5.00 × 10^−8^ m^3^ of sample and 1.95 × 10^−6^ m^3^ of DPPH solution at 0.06 mol/m^3^ (mol/v) (*v*/*v*). The DPPH remaining at the end of the reaction was determined and quantified, using the standard Trolox curve. The antioxidant activity of the DPPH method was expressed in micromol of Trolox equivalent per gram of sample on a dry basis (µmol TE/g d.b.).

### 2.4. Cell Culture

HepG2 cells (liver hepatocellular carcinoma) were cultured for the cytotoxicity and cytoprotection assays of the extracts. The HepG2 cell lines were stored under cryogenics in liquid nitrogen, added with fetal bovine serum and DMSO. The cells were thawed, centrifuged, and the supernatant was discarded (fetal bovine serum+DMSO). Then, the HepG2 cells were added with DMEM medium and fetal bovine serum, transferred to T75 flasks and incubated at 310.15 K and 5% CO_2_, under saturated humidity. The HepG2 cells contained in T75 flasks were washed twice with Hanks’ balanced salt solution to remove possible cellular metabolites. Then, the cells were added with trypsin and incubated for 300 s at 310.15 K and 5% CO_2_, under saturated humidity, to remove the cells adhered to the walls of the flask. DMEM was added to resolubilize the cells. For counting, 5.00 × 10^−8^ m^3^ of trypan blue and 5.00 × 10^−8^ m^3^ of cells in DMEM in eppendorf were added, homogenized and placed on a microscope slide. Cell counting was performed with the aid of an Inverted Microscope (Axiovert 200, Zeiss, Jena, Germany) and a manual cell counter (GT-08HM, Global Trade, Jaboticabal, Brazil).

### 2.5. In Vitro Evaluation of Cytotoxicity and Cytoprotection

The oil and ethanolic extract that had the highest levels lutein content were subjected to cytotoxicity and cytoprotection assay with HepG2 cells. The samples were solubilized in ethanol to a concentration of 8 kg/m^3^ (*m*/*v*). For cytotoxicity tests, the samples were solubilized in DMEM medium at five concentration points (0.20, 0.10, 0.05, 0.025, and 0.01 kg/m^3^) (*m*/*v*). For cytoprotection assays, the samples were solubilized in DMEM medium at a concentration of 0.05 kg/m^3^ and 1.00 × 10^−8^ m^3^ of hydrogen peroxide was added (*v*/*v*). The exposure was performed in 96-well microplates, where 1.00 × 10^−7^ m^3^ of HepG2 cells solubilized in DMEM medium (13:87) (*v*/*v*) were added, which was equivalent to a concentration of 6.70 × 10^3^ CFU/well. The microplates were incubated at 310.15 K and 5% CO_2_ under saturated humidity for 24 h. Then, 1.00 × 10^−7^ m^3^ of each sample concentration was added to the plates. The incubation times were 24 and 48 h for cytotoxicity tests and 24, 48 and 72 h for cytoprotection tests. For the colorimetric reaction, the MTT Assay Kit was used, using the Mosman protocol [47], where 1.00 × 10^−7^ m^3^ of the MTT solution were added to the wells and the microplates were incubated for 3 h at 310.15 K and 5% CO_2_, under saturated moisture in the absence of light. The MTT solution was removed and 1.00 × 10^−7^ m^3^ of DMSO was added, incubating the plates again for 1 h, under the same conditions. After the reaction, the microplates were read in an Absorbance Microplate Reader (Elx800, BioTek, Winooski, USA) at 570 nm, in duplicate. Cytotoxicity and cytoprotection were determined by the average percentage of cell survival in relation to the unexposed control.

### 2.6. Statistical Analysis

The means and standard deviations were calculated for all analyses. The results of the physical–chemical analyses were submitted to Tukey test, at a significance level of 5% (*p* < 0.05). The results of the cytotoxicity and cytoprotection tests were normalized and subjected to the ANOVA test and Tukey test, at a significance level of 5% (*p* < 0.05). Excel 2000 SR-1 (Microsoft, Troy, NY, USA), Statistica Kernel Release 7.1 (StartSoft Inc., Tulsa, OK, USA) and Bioestat (version 5.3) programs were used as tools.

## 3. Results and Discussion

### 3.1. Global Yield Isotherms

Global yield isotherms of extracts of *B. crassifolia* pulp obtained by CO_2_-SFE and CO_2_+EtOH-SFE at 343.15 K can be seen in Figure 1. The global yields of CO_2_-SFE ranged from 8.38% d.b. ± 0.13% d.b. (22 MPa-695 kg/m^3^) to 9.83% d.b. ± 0.36% d.b. (49 MPa-900 kg/m^3^). It can be seen that the increase in pressure and solvent density also increased the extraction yields. According to Silva et al. [48], pressure variations in isothermal conditions define the performance of an SFE system, since this parameter has a great influence on fluid hydrodynamics, solubility, and mass transfer. In the study by Pires et al. [13], for 323.15 K and 333.15 K isotherms of CO_2_-SFE of murici pulp, such behavior was also observed. When comparing the influence of temperature on this response obtained in both studies, it can be observed that global yields also increased with increasing temperature. This can be attributed to the increase in the solvation power of the solvents in the most drastic extraction conditions, which allowed a greater solubilization of the murici pulp compounds [49]. This behavior has also been observed by other authors [22].

Global yields of CO_2_+EtOH-SFE ranged from 5.47% d.b. ± 0.40% d.b. (22 MPa-775 kg/m^3^) to 9.77% d.b. ± 0.10% d.b. (49 MPa–944 kg/m^3^). It was possible to observe that the increase in pressure and density also increased the extraction yields. An inverse behavior was observed for 323.15 and 333.15 K isotherms, using the co-solvent in the dynamic period of CO_2_+EtOH-SFE of murici pulp [13]. This demonstrates that the use of co-solvent only in the static extraction period, at higher temperatures, made it possible to obtain an extract with high yield. This is due to the longer contact time between the solvent mixture and the solutes, which caused an increase in the solubilization of these compounds, with a consequent increase in these yields. Thus, it can be said that applying ethanol as a co-solvent (10% *v*/*v*), in the static period, increases the solubilization of polar bioactive compounds.

### 3.2. Lutein Content

Lutein content of the defatted pulps, oils, and ethanolic extracts of *B. crassifolia* pulp can be seen in Table 1. Lutein contents of oils ranged from 62.38 µg/g d.b. ± 0.67 µg/g d.b. (22 MPa-695 kg/m^3^) to 224.77 µg/g d.b. ± 0.67 µg/g d.b. (49 MPa-900 kg/m^3^). These values were higher than those obtained for the murici pulp extract obtained by maceration with acetone (43.90 µg/g) [4]. The increase in pressure increased the lutein levels in the oils. This behavior was also observed by Yen et al. [50], for lutein extraction with supercritical CO_2_. The high content of lutein in oils was mainly due to lutein being linked to fatty acids present in the plant matrices [51]. Thus, the nonpolar extract which presented the highest global yield consequently had the highest lutein content. The lutein composition of defatted pulps ranged from 21.93 µg/g d.b ± 0.10 µg/g d.b. (32 MPa-804 kg/m^3^) to 30.31 µg/g d.b. ± 0.06 µg/g d.b. (22 MPa-695 kg/m^3^). Pressure did not present a defined behavior for these results; however, the fact that the higher content of lutein in the defatted matrix was obtained in the condition of lower content in the oils is justified, since the residual lutein remained in the extraction bed. The lutein levels in ethanolic extracts ranged from 88.46 µg/g d.b. ± 0.58 µg/g d.b. (49 MPa-944 kg/m^3^) to 242.16 µg/g d.b. ± 0.55 µg/g d.b. (22 MPa-775 kg/m^3^). These results were superior to that found for the aqueous extract of murici pulp (23.39 µg/g) [52]. In this case, the increase in pressure reduced the levels of lutein, presenting an opposite behavior to that of CO_2_-SFE. This shows that lutein was not obtained in CO_2_-SFE, but in CO_2_+EtOH-SFE. It was also possible to observe that the use of ethanol as a co-solvent enabled a good recovery of the remaining lutein in defatted pulp. This behavior was also evidenced by Cobb et al. [53]. Although carotenoids are mostly nonpolar compounds, lutein has an intermediate polarity [26]. This explains the presence of this compound in similar amounts in both extracts. When comparing the levels of lutein in the oil and ethanol extract of murici pulp obtained at 343.15 K in the present study with those found in the study by Pires et al. [13] obtained at 323.15 and 333.15 K, it is possible to observe that the increase in the process temperature increased the lutein levels in supercritical extractions with CO_2_ and with CO_2_+ethanol. This may have occurred due to the increase in the solute vapor pressure, which increased the diffusion rate and promoted an increase in the transfer rate of mass of lutein contained in the plant matrix for supercritical solvents [49]. This demonstrates that better recovery of lutein from murici pulp occurs in supercritical extraction at 343.15 K.

### 3.3. Phenolic Compounds

The contents of phenolic compounds in the defatted pulps and ethanolic extracts of *B. crassifolia* pulp can be seen in Table 1. For the recovery of phenolic compounds, it is important to emphasize that a pretreatment with supercritical CO_2_ is necessary to obtain an extraction bed more concentrated in polar bioactive compounds [54]. This explains the fact that phenolic compounds from murici oils were not detected, containing results below the detection limit of the analysis (LOD = 1.22 × 10^−4^ kg/m^3^), which indicates that these compounds were concentrated in defatted beds. This behavior is quite consistent, since phenolic compounds are polar substances and supercritical CO_2_ is a nonpolar solvent [55]. Therefore, defatted beds were used to extract these compounds. The levels of phenolic compounds in the defatted pulps were in the range of 12.02 mg GAE/g d.b. ± 0.42 mg GAE/g d.b. (49 MPa-900 kg/m^3^) and 24.58 mg GAE/g d.b. ± 0.86 mg GAE/g d.b. (22 MPa-695 kg/m^3^). The increase in pressure decreased the levels of phenolic compounds in defatted pulps. This behavior was also observed by Batista et al. [23] for defatted açaí pulp (*Euterpe oleracea*), obtained under the same extraction conditions as the present study. This was possibly due to the fact that the increased pressure facilitated the enzymatic oxidation of defatted pulps [56]. The levels of phenolic compounds in ethanolic extracts ranged from 6.73 mg GAE/g d.b. ± 0.15 mg GAE/g d.b. (22 MPa-775 kg/m^3^) to 20.63 mg GAE/g d.b. ± 0.76 mg GAE/g d.b. (49 MPa-944 kg/m^3^). These values were higher than those found for the murici pulp extract obtained with an extracting solution of methanol, ethanol, distilled water, and hydrochloric acid (69:20:10:1, *v*/*v*/*v*/*v*) (0.80 mg GAE/g) [57]. The increase in pressure also increased the levels of phenolic compounds in ethanolic extracts. Some studies have also reported this behavior [58,59]. This may have occurred because the increased pressure promoted the rupture of plant tissues, cell walls, and organelles, improving the mass transfer of the solvent to the sample, and of compounds to the solvent [60,61].

### 3.4. Flavonoid Content

Flavonoid contents in the defatted pulps and ethanolic extracts of *B. crassifolia* pulp are shown in Table 1. The flavonoid contents of murici oils were not detected, being below the detection limit of the analysis (LOD = 1.28 × 10^−4^ kg/m^3^). This behavior was similar to that obtained for the content of phenolic compounds, since flavonoids are one of the classes belonging to the group of polyphenols [62]. Flavonoid levels in defatted pulps ranged from 0.34 mg QE/g d.b. ± 0.01 mg QE/g d.b. (49 MPa-900 kg/m^3^) to 0.52 mg QE/g d.b. ± 0.01 mg QE/g d.b. (32 MPa-804 kg/m^3^). The flavonoid content did not show a defined linear behavior with increasing pressure. This behavior demonstrates that the selectivity of these compounds is not defined mainly by pressure, but by the vapor pressure of solutes. The levels of flavonoids in ethanolic extracts were in the range of 0.59 mg QE/g d.b. ± 0.02 mg QE/g d.b. (22 MPa-775 kg/m^3^) to 0.65 mg QE/g d.b. ± 0.07 mg QE/g d.b. (32 MPa-858 kg/m^3^); however, there was no significant difference between these values. These results were higher than that found for the methanolic extract of murici pulp (0.2 mg QE/g) [63]. Although there was no difference between the results, it was possible to observe that the levels of recovery/concentration of flavonoids in the extracts were different, being 1.45, 1.01, and 1.85 times for pressures of 22, 32, and 49 MPa, respectively. It can be seen that up to 32 MPa, the increase in pressure reduced the recovery of flavonoid content. This may have been due to repulsive solute-solvent interactions. This behavior was also observed by Chauhan et al. [64] for the extraction of flavonoids from black grape juice (*Vitis vinifera*), at high pressures. However, ethanolic extract obtained at 49 MPa was the one that showed the best recovery/concentration of these compounds in comparison to the other extraction conditions, since its flavonoid content almost doubled (0.64 mg QE/g d.b.) in relation to the defatted pulps content used in the CO_2_+EtOH-SFE, in this same condition (0.34 mg QE/g d.b.). This was possibly due to the increase in density of the solvent mixture that allowed a greater transfer of these compounds to the extract. Therefore, it can be said that 49 MPa pressure facilitated the extraction of flavonoids of murici pulp.

### 3.5. Fatty Acids and Functional Quality

The fatty acid profile of oils of *B. crassifolia* pulp is shown in Figure 2 and Table 2. In all pressures evaluated, it was possible to observe that there was no significant difference among the fatty acid profiles, with the majority being oleic acid, followed by palmitic and linoleic acids. This behavior and fatty acid profile were also obtained for the extraction of murici pulp conducted at temperatures from 323.15 K to 333.15 K, and pressures from 15 MPa to 42 MPa [13]. This allows us to affirm that temperature and pressure did not influence the composition of fatty acids present in the oils of murici pulp. In addition, the fatty acid profiles of the oils presented about 63% of unsaturated fatty acids (UFA), being predominantly monounsaturated with about 40% (MUFA). This was confirmed by the results obtained from the saturated fatty acids/unsaturated fatty acids (SFA/UFA) relation, which was 0.60. However, the fatty acid profile of murici pulp oil obtained with chloroform-methanol 2:1 (*v*/*v*) was slightly different from that observed, with palmitic, oleic, and linoleic acid as major factors [65]. These results indicate that the use of conventional extractions makes it possible to obtain murici pulp oils that are richer in saturated fatty acids, which reduces their functional potential. It can be observed that the atherogenicity index (AI) and thrombogenicity index (TI) were relatively low, whereas the hypocholesterolemic index (HI) was high. This demonstrates that the extracts have functional potential since the values of these indices indicate anti-atherosclerogenic, antithrombogenic, and antihypercholesterolemic effects [11,14]. These same functional potentials were observed in uxi oil [66] and bacaba oil [14] obtained by supercritical extraction. Therefore, this confirms that in any reported supercritical extraction condition, it is possible to obtain a relatively stable nonpolar extract of murici pulp with antithrombogenic, antihypercholesterolemic, and anti-atherosclerogenic activities, properties, constituting a product with functional quality.

### 3.6. Probable Triglyceride Composition

The composition of triglycerides of oils of *B. crassifolia* pulp is shown in Table 3. For the prediction of this composition, palmitic, oleic, linoleic, palmitoleic, and stearic acids were used, since they represented about 99% of the total fatty acid composition of oils of *B. crassifolia* pulp. From these acids, it was possible to obtain 89% of the triglyceride composition present in oils. It can be seen that for all extraction conditions, the major triglycerides were POLi (palmitic, oleic, and linoleic), POO (palmitic, oleic, and oleic), PPO (palmitic, palmitic, and oleic), and LiOO (linoleic, oleic, and oleic), constituting 58% of the triglyceride profile. These results are in accordance with the fatty acid profiles obtained in the present study, since the major triglycerides obtained were formed by palmitic, oleic, and linoleic acids. The use of triglyceride profile prediction has also been reported by other authors and the compositions obtained are quite similar to those obtained experimentally [22,66,67]. All triglycerides identified can be classified as long-chain triglycerides (LCT), since they have more than 36 carbon atoms. It is important to highlight that LCT have been used in several applications, such as in short and long-term treatments with ketogenic diet on cortical spreading depression [68], to increase bioavailability and reduce hepatic metabolism of neurotropic agents [69], to reduce the release of inflammatory mediators from the gastrointestinal tract, and exert a protective effect on the inflammatory response of liver during sepsis [18]. Therefore, the knowledge of the triglyceride composition in the extracts is of great importance for its use in food and pharmaceutical industries.

### 3.7. Oxygen Radical Absorbance Capacity (ORAC)

ORACs of the defatted pulps, oils and ethanolic extracts of *B. crassifolia* pulp are shown in Table 1. ORAC values of oils ranged from 32.83 µmol TE/g d.b. ± 0.27 µmol TE/g d.b. (49 MPa-900 kg/m^3^) to 43.48 µmol TE/g d.b. ± 0.88 µmol TE/g d.b. (32 MPa-804 kg/m^3^). It can be observed that the increase in pressure also increased the antioxidant capacity up to 32 MPa, reducing it again at 49 MPa. This behavior was also observed for oils of murici pulp obtained under the same CO_2_ densities, but different conditions of temperature and pressure [13]. This allows us to say that the bioactive compounds that confer the antioxidant capacity to these extracts are more soluble at 800 kg/m^3^, which confirms the selectivity of supercritical CO_2_. The ORAC results of defatted pulps ranged from 1.45 µmol TE/g d.b. ± 0.03 µmol TE/g d.b. (32 MPa-804 kg/m^3^) to 1.90 µmol TE/g d.b. ± 0.06 µmol TE/g d.b. (22 MPa-695 kg/m^3^). The condition of highest ORAC of CO_2_-SFE extracts, consequently, was the one with the lowest ORAC of the defatted pulps, since more compounds were removed from the extraction bed. The ORAC results of ethanolic extracts ranged from 100.88 µmol TE/g d.b. ± 1.41 µmol TE/g d.b. (22 MPa-775 kg/m^3^) to 122.61 µmol TE/g d.b. ± 3.79 µmol TE/g d.b. (49 MPa-944 kg/m^3^). These results were superior to that found for the murici pulp extract obtained with methanol/water solution (80:20, *v*/*v*) (26.50 µmol TE/g d.b.) [63]. It was possible to observe that the increase in pressure and, consequently, in the mixture density, also increased the antioxidant capacity of extracts. It can also be noted that the antioxidant capacities of the extracts obtained with ethanol as co-solvent were much higher than those that used only CO_2_. This behavior was also observed by Serra et al. [54] when using ethanol as co-solvent (proportion 90:10, *v*/*v*). This occurred due to the change in polarity of the solvent, which allowed the extraction of polar bioactive substances, such as phenolic compounds and flavonoids.

### 3.8. Antioxidant Capacity (DPPH)

The results of DPPH of the defatted pulps, oils and ethanolic extracts of *B. crassifolia* pulp can be seen in Table 1. DPPH values of oils ranged from 6.01 µmol TE/g d.b. ± 0.31 µmol TE/g d.b. (22 MPa-695 kg/m^3^ and 49 MPa-994 kg/m^3^) to 6.04 µmol TE/g d.b. ± 0.19 µmol TE/g d.b. (32 MPa-804 kg/m^3^), but there was no significant difference between them, suggesting that the pressure did not influence the antioxidant activity of nonpolar extracts. The values of DPPH of the defatted pulps ranged from 1.37 µmol TE/g d.b. ± 0.13 µmol TE/g d.b. (49 MPa-900 kg/m^3^) to 2.47 µmol TE/g d.b. ± 0.10 µmol TE/g d.b. (22 MPa-695 kg/m^3^). It can be seen that the increase in pressure decreased the antioxidant activity of the defatted pulps. DPPH values of ethanolic extracts ranged from 12.87 µmol TE/g d.b. ± 0.38 µmol TE/g d.b. (22 MPa-775 kg/m^3^) to 17.14 µmol TE/g d.b. ± 0.45 µmol TE/g d.b. (49 MPa-994 kg/m^3^). These results were superior to those found for the hydrophilic extracts of murici pulp obtained with acetone/water/acetic acid (70:29:5:0.5, *v*/*v*/*v*) (3.73 µmol TE/g) [70]. Then, it was observed, one more time, that the increase in pressure also increased the antioxidant capacity of these extracts. This was similar to the behavior observed for ORAC. It is important to note that the methods used to determine the antioxidant action of murici extracts, in the present study, were obtained by different techniques and, therefore, cannot be compared with each other [71]. However, both presented satisfactory values, which allows us to say that the extracts are good sources of antioxidant compounds.

### 3.9. In Vitro Evaluation of Cytotoxicity and Cytoprotection

The oil and ethanolic extract of the *B. crassifolia* pulp used in the cytotoxicity and cytoprotection tests were obtained at 49 MPa-900 kg/m^3^ and 22 MPa-775 kg/m^3^, respectively. The percentages of cytotoxicity and cytoprotection are shown in Figure 3. It can be observed in the cytotoxicity tests of the oil that the levels survival rates ranged from 91.72% to 103.32% (±8.95%) in 24 h, and 101.08% to 113.57% (±8.26%) in 48 h. For the ethanolic extract, the ranges from 97.66% to 120.68% (±20.47%) were obtained in 24 h, and 94.27% to 111.60% (±10.79%) in 48 h. For both samples, it was observed that regardless of the concentration and incubation time, no significant differences were observed in the levels of cell viability in relation to the negative controls and ethanol, which demonstrates that no extract showed cytotoxicity in the concentration conditions addressed.

Murici oil cytoprotection assays showed survival values in the 90.34% to 112.81% (±9.60%) range in 24 h, 95.29% to 109.77% (±6.28%) in 48 h, and 95.21% to 106.18% (±4.73%) in 72 h. Significant differences were observed only at 72 h in relation to the levels of survival between the oil and the H_2_O_2_ control, where the survival levels of cells exposed with murici oil increased 19.60% in relation to the H_2_O_2_ control. This allows us to say that the oil presented a cytoprotective effect in 72 h of incubation, where the increased exposure time stimulated proliferation and inhibited cell apoptosis, being effective in neutralizing the oxidative stress induced by H_2_O_2_ in HepG2 cells. The cytoprotection result of murici oil suggests that compounds such as unsaturated fatty acids, present in the oil, were responsible for increasing the cytoprotection time under the studied conditions. According to Bak, Jun and Jeong [27], non-polar bioactive compounds have stood out for biological roles due to their greater bioavailability compared to polar bioactive compounds. This may have occurred due to the induction of significant changes in the composition of cellular fatty acids, with an increase in the levels of essential fatty acids, indicating a process of absorption of these important bioactive components [72]. The cytoprotection tests of the ethanolic extract showed survival values in the ranges of 96.30% to 103.44% (±3.12%) in 24 h, 79.74% to 85.25% (±2.45%) in 48 h and 85.79% to 97.21% (±5.08%) in 72 h. It was observed that regardless of the concentration and the incubation time, no significant differences were observed in the levels of cell viability in relation to the controls, which demonstrates that the ethanolic extract did not present cytoprotection using the sample concentration of 0.05 kg/m^3^ and the H_2_O_2_ concentration of 1.00 × 10^−8^ m^3^. These results indicate that, although the ethanolic extract has shown higher ORAC and DPPH results, attributed to the phenolic compounds, flavonoids and lutein, the sample concentration and the hydrogen peroxide concentration used was not sufficient to cause a significant increase in the levels of survival. An opposite behavior was reported by Barbosa et al. [28] when using the same concentrations of H_2_O_2_ and sample to evaluate the cytoprotective effect of aqueous extracts of *Pleurotus ostreatus* in HepG2 cells, where significant differences in survival levels were found. Therefore, it is suggested that future work be carried out exploring the gradients of sample concentration and hydrogen peroxide to adapt the cellular model of oxidative stress to ethanol extracts of murici. Therefore, the non-toxic effects of both extracts and the cytoprotective effect of the oil suggest that both the oil and the ethanolic extract of the *B. crassifolia* pulp can be used for the development of food, cosmetic and pharmaceutical products.

## 4. Conclusions

Oils showed good levels of lutein, antioxidant capacity, and unsaturated fatty acids.

It was observed that the fatty acid compositions did not vary with the operating conditions of extraction. Therefore, in any condition of supercritical extraction reported, it was possible to obtain relatively stable oils of murici with antithrombogenic, antihypercholesterolemic, and anti-atherosclerogenic activities, constituting a product with functional quality.

The fatty acid profile allowed the prediction of triglyceride composition and demonstrated that oils are constituted of long-chain triglycerides, which can guide possible applications in food and pharmaceuticals sectors.

The use of co-solvent only in the static period made it possible to obtain polar extracts containing lutein, phenolic compounds, flavonoids, and antioxidant activity.

Obtaining the sequential extraction of murici pulp (*B. crassifolia*) by supercritical extraction at 343.15 K enabled the valuation and better use of the plant matrix for the production of high added-value extracts, with different compositions, and wide industrial applications.

Both extracts did not show cytotoxicity and only murici oil showed a cytoprotective effect in 72 h of exposure, where the increased exposure time stimulated proliferation and inhibited cell apoptosis, being effective in neutralizing the oxidative stress induced by H_2_O_2_ in HepG2 cells.

The results of this study showed remarkable nutritional/nutraceutical value of murici pulp extracts (*B. crassifolia*), and qualify it as a potential resource for use as food and in the development of dietary supplements.

## Figures and Tables

**Figure 1 foods-10-00737-f001:**
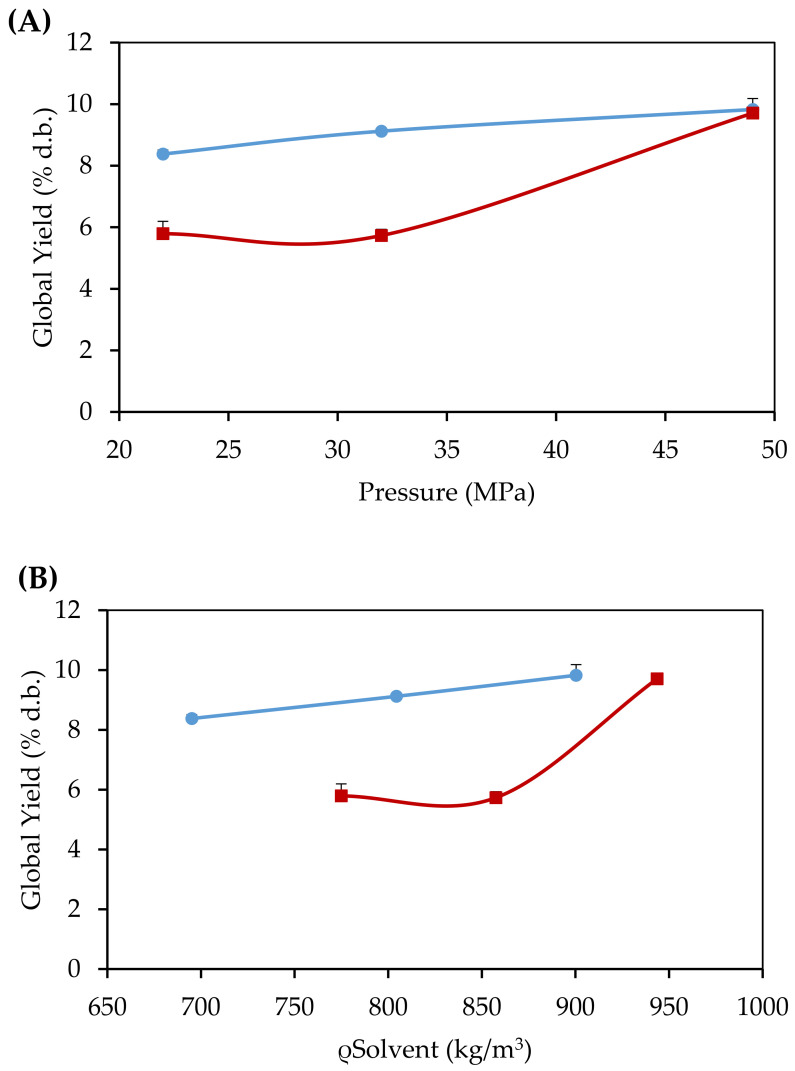
Global yield isotherms of murici pulp extracts (*B. crassifolia*) obtained by CO_2_-SFE (
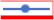
) (**A**) and CO_2_+EtOH-SFE (
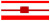
) (**B**), at 343.15 K (Standard deviations ≤ 0.4%).

**Figure 2 foods-10-00737-f002:**
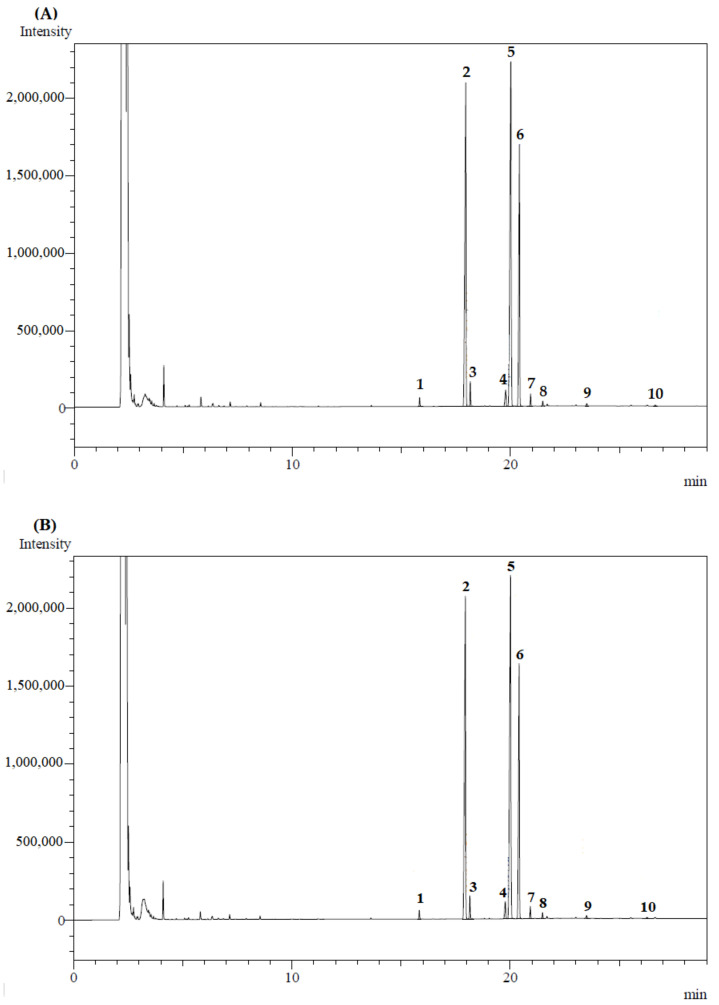
Total ion chromatogram of oils of murici pulp extracts (*B. crassifolia*) obtained by CO_2_-SFE, at 343.15 K and 22 MPa (**A**), 32 MPa (**B**), and 49 MPa (**C**), with the peak numbers corresponding to the compounds identification cited in Table 2.

**Figure 3 foods-10-00737-f003:**
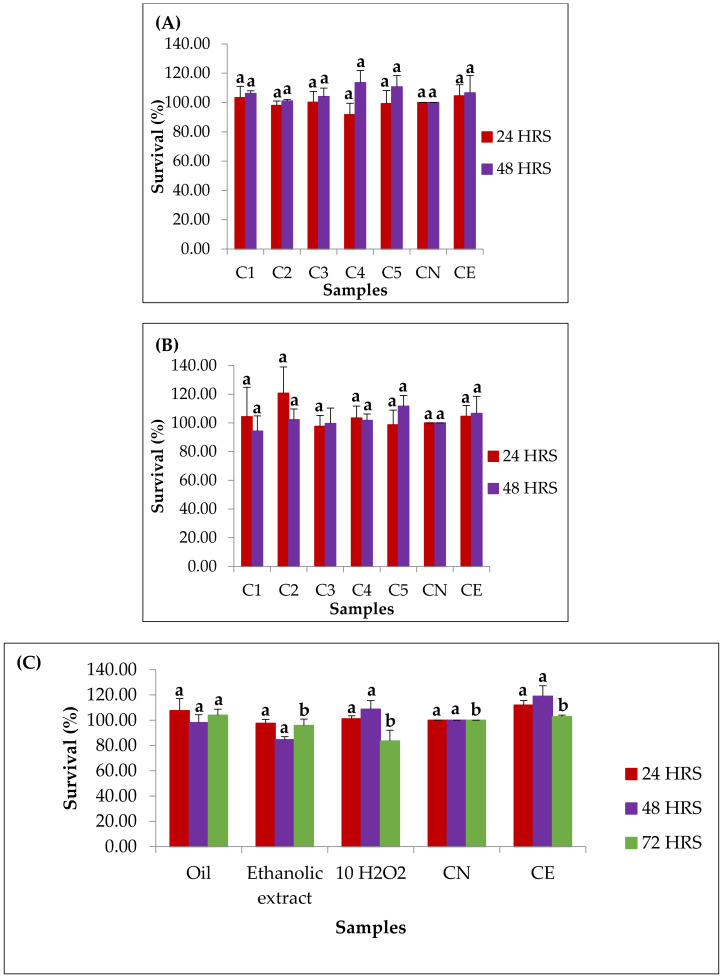
Cytotoxicity data of oil (**A**) and ethanolic extract (**B**) of murici pulp (*B. crassifolia*) obtained by CO_2_-SFE and CO_2_+EtOH-SFE, together with cytoprotection data using 0.05 kg/m^3^ of samples in HepG2 cells treated with H_2_O_2_ (**C**). * Different letters for the same incubation time showed a difference in significance level of 5% (*p* < 0.05) (ANOVA; Tukey test for multiple comparisons); C1: concentration 0.20 kg/m^3^; C2: concentration 0.10 kg/m^3^; C3: concentration 0.05 kg/m^3^; C4: concentration 0.025 kg/m^3^; C5: concentration 0.01 kg/m^3^; CN: negative control; CE: ethanol control; 10 H_2_O_2_: peroxide control.

**Table 1 foods-10-00737-t001:** Operating conditions of CO_2_–SFE and CO_2_+EtOH–SFE of murici pulp (*B. crassifolia*) and characterization of the different defatted pulps and extracts.

Samples	Pressure (MPa)	ρSolvent (kg/m^3^)	Luteín Content (µg/g d.b.) *	Phenolic Compounds (mg GAE/g d.b.) *	Flavonoids Content (mg QE/g d.b.) *	ORAC (µmol TE/g d.b.) *	DPPH (µmol TE/g d.b.) *
Oils (CO_2_ –SFE)	22	695	62.38 ± 0.67 ^c^	n.d.	n.d.	34.44 ± 0.21 ^b^	6.01 ± 0.31 ^a^
	32	804	196.18 ± 0.90 ^b^	n.d.	n.d.	43.48 ± 0.88 ^a^	6.04 ± 0.19 ^a^
	49	900	224.77 ± 0.67 ^a^	n.d.	n.d.	32.83 ± 0.27 ^c^	6.01 ± 0.19 ^a^
Defatted pulps	22	695	30.31 ± 0.06 ^a^	24.58 ± 0.86 ^a^	0.43 ± 0.01 ^b^	1.90 ± 0.06 ^a^	2.47 ± 0.10 ^a^
	32	804	21.93 ± 0.10 ^c^	19.67 ± 0.27 ^b^	0.52 ± 0.01 ^a^	1.45 ± 0.03 ^c^	1.69 ± 0.15 ^b^
	49	900	22.58 ± 0.10 ^b^	12.02 ± 0.42 ^c^	0.34 ± 0.01 ^c^	1.58 ± 0.04 ^b^	1.37 ± 0.13 ^c^
Ethanolic extracts (CO_2_+EtOH–SFE)	22	775	242.16 ± 0.55 ^a^	6.73 ± 0.15 ^c^	0.65 ± 0.07 ^a^	100.88 ± 1.41 ^b^	12.87 ± 0.38 ^c^
	32	858	163.76 ± 0.94 ^b^	7.93 ± 0.27 ^b^	0.59 ± 0.02 ^a^	117.45 ± 2.40 ^a^	15.01 ± 0.19 ^b^
	49	944	88.46 ± 0.58 ^c^	20.63 ± 0.76 ^a^	0.64 ± 0.01 ^a^	122.61 ± 3.79 ^a^	17.14 ± 0.45 ^a^

* Different letters in the same column, per sample, showed a difference in significance level of 5% (*p* < 0.05); ORAC: oxygen radical absorbance capacity; DPPH: Antioxidant Activity; CO_2_ –SFE: extraction with supercritical CO_2_; CO_2_+EtOH–SFE: extraction with supercritical CO_2_ and ethanol; n.d: not detected; LOD of lutein = 6.17 × 10^−4^ kg/m^3^; LOD of phenolic compounds = 1.22 × 10^−4^ kg/m^3^; LOD of flavonoids = 1.28 × 10^−4^ kg/m^3^; LOD of ORAC = 5.92 × 10^−4^ mol/m^3^; LOD of DPPH = 6.46 × 10^−4^ mol/m^3^.

**Table 2 foods-10-00737-t002:** Total fatty acids and functional potential of the oils of murici pulp (*B. crassifolia*) obtained by CO_2_-SFE at 343.15 K.

Peak Number		Relative Area (% d.b.) *		
	Fatty Acids	22 MPa- 695 kg/m^3^	32 MPa- 804 kg/m^3^	49 MPa- 900 kg/m^3^
1	C14:0	0.51 ^a^	0.51 ^a^	0.52 ^a^
2	C16:0	34.18 ^a^	34.38 ^a^	34.39 ^a^
3	C16:1	1.43 ^a^	1.39 ^a^	1.39 ^a^
4	C18:0	1.57 ^a^	1.66 ^a^	1.67 ^a^
5	C18:1	39.21 ^a^	39.04 ^a^	39.04 ^a^
6	C18:2	21.78 ^a^	21.58 ^a^	21.53 ^a^
7	C18:3	0.72 ^a^	0.72 ^a^	0.74 ^a^
8	C20:0	0.30 ^a^	0.36 ^a^	0.37 ^a^
9	C22:0	0.18 ^a^	0.23 ^a^	0.24 ^a^
10	C24:0	0.11 ^a^	0.12 ^a^	0.13 ^a^
-	SFA	36.85 ^a^	37.26 ^a^	37.32 ^a^
-	UFA	63.14 ^a^	62.73 ^a^	62.70 ^a^
-	MUFA	40.64 ^a^	40.43 ^a^	40.43 ^a^
-	PUFA	22.50 ^a^	22.30 ^a^	22.27 ^a^
-	S/U	0.58 ^a^	0.59 ^a^	0.60 ^a^
-	AI	0.57 ^a^	0.58 ^a^	0.58 ^a^
-	IT	1.09 ^a^	1.10 ^a^	1.10 ^a^
-	HH	1.82 ^a^	1.80 ^a^	1.80 ^a^

* The results were obtained by mass basis; C14:0 (myristic acid); C16:0 (palmitic acid); C16:1 (palmitoleic acid); C18:0 (stearic acid); C18:1 (oleic acid); C18:2 (linoleic acid); C18:3 (linolenic acid); C20:0 (arachidic acid); C22:0 (behenic acid); C24:0 (lignoceric acid); SFA (saturated fatty acids); UFA (unsaturated fatty acids); MUFA (monounsaturated fatty acids); PUFA (polyunsaturated fatty acids); AI (atherogenicity index); TI (thrombogenicity index); HI (hypocholesterolemic index). The standard deviations for all fatty acids were lower than 1.8%. Different letters in the same line showed a difference in significance level of 5% (*p* < 0.05).

**Table 3 foods-10-00737-t003:** Prediction of triglyceride composition of the oils of murici pulp (*B. crassifolia*) obtained by CO_2_-SFE at 343.15 K.

Triglycerides	X:Y *	MM (g/mol)	Fração Molar (% d.b.)		
			22 MPa- 695 kg/m^3^	32 MPa- 804 kg/m^3^	49 MPa- 900 kg/m^3^
**PPP**	48:0	806	3.63	4.05	4.07
**OOO**	54:3	884	6.45	5.98	5.99
**LiLiLi**	54:6	878	1.14	1.02	1.01
**PPO**	50:1	832	13.18	13.83	13.88
**POO**	52:2	858	15.97	15.76	15.79
**LiPP**	50:2	830	7.40	7.68	7.68
**PLiLi**	52:4	854	5.03	4.86	4.83
**LiOO**	54:4	882	10.87	9.97	9.93
**OLiLi**	54:5	880	6.10	5.54	5.49
**POLi**	52:3	856	17.93	17.50	17.46
**OPaP**	50:2	830	1.17	1.14	1.13
**PSO**	52:1	860	1.05	1.32	1.33

* X = Number of carbons; Y = Number of double bonds; MM = molar mass; P (palmitic acid); O (oleic acid); Li (linoleic acid); Pa (palmitoleic acid); S (stearic acid).

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
