# Peer review of "Bioactive Compounds and Evaluation of Antioxidant, Cytotoxic and Cytoprotective Effects of Murici Pulp Extracts (Byrsonima crassifolia) Obtained by Supercritical Extraction in HepG2 Cells Treated with H2O2"

_foods, 2021, doi:10.3390/foods10040737_

Round 1

Reviewer 1 Report

Title: Bioactive compounds and evaluation of antioxidant, cytotoxic and cytoprotective effects of murici pulp extracts (Byrsonima crassifolia) obtained by supercritical extraction in HepG2 cells treated with H2O2

Journal: Foods

foods-1112083

The manuscript aims with the characterization of bioactive compounds and the evaluation of the antioxidant potential in oil and ethanolic extract obtained by two-step supercritical extraction. The strength of this paper is its novelty. There are only a few number of studies (30 original research papers in WOS when keywords “murici pulp” was used) in the literature focused in these Brazilian fruit. The present manuscript shows for the first time the effect of different processing factors on global yield, chemical composition and bioactivity of murici extracts. General objective is clear and the experimental approach was well designed to address the objective. Starting material used has been well defined and characterized.

Methodology was described in detail and in general is adequate to achieve the objective. Nevertheless, I have a few comments at this point. The authors should indicate which solvents were used to prepare sample solutions and concentrations prepared to determine the phenolic compounds, flavonoids, fatty acids, ORAC and DPPH analysis.

Results section is well organized and exposed in logical order. In general, interpretation of results is correct with the exception of cytoprotection experiments against oxidative stress. Figure 3c shows that viability of cells exposed to H2O2 for 24 and 48 h was similar compared to non-treated cells (negative control). That may indicate that the concentration of H2O2 used in the experiments was relatively low to observe a significant reduction in cell viability in a timeframe of 24-48 h. Therefore, the oxidative stress cellular model used to perform this sort of experiments was not adequate to study the antioxidant potential of murici pulp extracts. My recommendation is to include results from new experiments in which authors will draw more appropiate conclusions on the cytoprotective effects of murici extracts against oxidative stress. According to this comment, results, discussion and conclusions should be revised.

Statistical analysis should be included in Figure 3. ANOVA and post-hoc test for mean comparison between the different treatment groups have to be included. In Figure 3c caption authors should indicate sample dosis used in cytoprotection experiments.

Data used be expressed using the units of the International System throughout the manuscript (e.g. temperature units should be changed to degrees Celsius).

Author Response

Dear Reviewer, I would like to thanks for all suggestions in comments! The responses were attached !

Best Regards!

Reviewer 2 Report

The manuscript entitled “Bioactive compounds and evaluation of antioxidant, cytotoxic and cytoprotective effects of murici pulp extracts (Byrsonima crassifolia) obtained by supercritical extraction in HepG2 cells treated with H2O2” authored by Flávia Cristina Seabra Pires and colleagues, deals with the investigation of the phytochemical profile and antioxidant, cytotoxic and cytoprotective properties of extracts from Byrsonima crassifolia pulp obtained supercritical extraction on HepG2 cells stressed with H2O2. The article is well written, contains very interesting data. However, several typos are contained in the text, and some changes must be made.

  1. The abstract should be rewritten, because it is missing of the current state of art and aims of the study. It simply report the obtained results. Please, modify it accordingly to journal instructions.
  2. Keywords should be words not contained within the title of the manuscript. Since many of these words are present in the title, you strongly advise to modify and replace them.
  3. Why is table 1 reported in Materials and Methods section? Since It contains results, it should be moved in Results and Discussion Section.
  4. The quality of all figures should be generally improved. Since Foods does not apply any additional charge for publishing colour images, I strongly suggest to remake the figure using different colours for each conditions. This would certainly increase the quality of the figures. Moreover, panel letters should be reported in UPPERCASE and in bold in the upper left corner of each panel.
  5. Authors should check the manuscript and correct several typos contained all over the main text.
  6. I understand that the authors do not have the possibility to evaluate the phytochemical profile of the extract, however what is known about the chemical characterization of essential oils from Byrsonima crassifolia obtained by classical extraction methods or SFE should be reported, both in the introduction and in the results and discussion sections. Authors should correlate their experimental results (flavonols, polyphenols, etc.) with previously reported data.
  7. In Table 1, the authors should replace the “-“ with "n.d." (not detected). In addition, authors should provide information on why it was not possible to provide a value for those samples (authors should provide data related to limit of detection (LOD) for these assays).
  8. Section 2.3.5. is not clear. Please, provide more information.

Author Response

(The authors gave the same response as above.)

Round 2

Reviewer 1 Report

Authors have addressed correctly all comments of the review report. It can be accepted in its current form.